# Classifier-to-Generator Attack: Estimation of Training Data Distribution from Classifier

## Abstract

Suppose a deep classification model is trained with samples that need to be kept private for privacy or confidentiality reasons. In this setting, can an adversary obtain the private samples if the classification model is given to the adversary? We call this reverse engineering against the classification model the *Classifier-to-Generator (C2G) Attack*. This situation arises when the classification model is embedded into mobile devices for offline prediction (e.g., object recognition for the automatic driving car and face recognition for mobile phone authentication).

For C2G attack, we introduce a novel GAN, *PreImageGAN*. In PreImageGAN, the generator is designed to estimate the the sample distribution conditioned by the preimage of classification model $f$, $P(X|f(X) = y)$, where $X$ is the random variable on the sample space and $y$ is the probability vector representing the target label arbitrary specified by the adversary. In experiments, we demonstrate PreImageGAN works successfully with hand-written character recognition and face recognition. In character recognition, we show that, given a recognition model of hand-written digits, PreImageGAN allows the adversary to extract alphabet letter images without knowing that the model is built for alphabet letter images. In face recognition, we show that, when an adversary obtains a face recognition model for a set of individuals, PreImageGAN allows the adversary to extract face images of specific individuals contained in the set, even when the adversary has no knowledge of the face of the individuals.

## 1 Introduction

Recent rapid advances in deep learning technologies are expected to promote the application of deep learning to online services with recognition of complex objects. Let us consider the face recognition task as an example. The probabilistic classification model $f$ takes a face image $\mathbf{x}$ and the model predicts the probability of which the given face image is associated with an individual $t$, $f(\mathbf{x}) \simeq \Pr[T = t | X = \mathbf{x}]$.

The following three scenarios pose situations that probabilistic classification models need to revealed in public for online services in real applications:

**Prediction with cloud environment:** Suppose an enterprise provides an online prediction service with a cloud environment, in which the service takes input from a user and returns predictions to the user in an online manner. The enterprise needs to deploy the model $f$ into the cloud to achieve this.

**Prediction with private information:** Suppose an enterprise develops a prediction model $f$ (e.g., disease risk prediction) and a user wishes to have a prediction of the model with private input (e.g., personal genetic information). The most straightforward way to preserve the user's privacy entirely is to let the user download the entire model and perform prediction on the user side locally.

**Offline prediction:** Automatic driving cars or laptops with face authentication contain face/object recognition systems in the device. Since these devices are for mobile use and need to work standalone, the full model $f$ needs to be embedded in the device.

In such situations that classification model $f$ is revealed, we consider a reverse-engineering problem of models with deep architectures. Let $\mathcal{D}_{\text{tr}}$ and $d_{X,T}$ be a set of training samples and its underlying distribution, respectively. Let $f$ be a model trained with $\mathcal{D}_{\text{tr}}$. In this situation, is it possible for an

adversary to obtain the training samples $\mathcal{D}_{tr}$ (or its underlying distribution $d_{X,T}$) if the classification model is given to the adversary?. If this is possible, this can cause serious problems, particularly when $\mathcal{D}_{tr}$ or $d_{X,T}$ is private or confidential information.

**Privacy violation by releasing face authentication:** Let us consider the face authentication task as an example again. Suppose an adversary is given the classification model $f$. The adversary aims to estimate the data (face) distribution of a target individual $t^*$, $d_{X|T=t^*}$. If this kind of reverse-engineering works successfully, serious privacy violation arises because individual faces are private information. Furthermore, once $d_{X|T=t^*}$ is revealed, the adversary can draw samples from $d_{X|T=t^*}$, which would cause another privacy violation (say, the adversary can draw an arbitrary number of the target's face images).

**Confidential information leakage by releasing object recognizer:** Let us consider an object recognition system for automatic driving cars. Suppose a model $f$ takes as input images from car-mounted cameras and detect various objects such as traffic signs or traffic lights. Given $f$, the reverse engineering reveals the sample distribution of the training samples, which might help adversaries having malicious intentions. For example, generation of adversarial examples that make the recognition system confuse without being detected would be possible. Also, this kind of attack allows exposure of hidden functionalities for privileged users or unexpected vulnerabilities of the system.

If this kind of attack is possible, it indicates that careful treatment is needed before releasing model $f$ in public considering that publication of $f$ might cause serious problems as listed above. We name this type of reverse engineering *classifier-to-generator (C2G) attack* . In principle, estimation of labeled sample distributions from a classification/recognition model of complex objects (e.g., face images) is a difficult task because of the following two reasons. First, estimation of generative models of complex objects is believed to be a challenging problem itself. Second, model $f$ often does not contain sufficient information to estimate the generative model of samples. In supervised classification, the label space is always much more abstract than the sample space. The classification model thus makes use of only a limited amount of information in the sample space that is sufficient to classify objects into the abstract label space. In this sense, it is difficult to estimate the sample distribution given only classification model $f$.

To resolve the first difficulty, we employ Generative Adversarial Networks (GANs). GANs are a neural network architecture for generative models which has developed dramatically in the field of deep learning. Also, we exploit one remarkable property of GANs, the ability to interpolate latent variables of inputs. With this interpolation, GANs can generate samples (say, images) that are not included in the training samples, but realistic samples[1].

Even with this powerful generation ability of GANs, it is difficult to resolve the second difficulty. To overcome this for the C2G attack, we assume that the adversary can make use of *unlabeled* auxiliary samples $\mathcal{D}_{aux}$ as background knowledge. Suppose $f$ be a face recognition model that recognizes Alice and Bob, and the adversary tries to extract Alice's face image from $f$. It is natural to suppose that the adversary can use public face image samples that do not contain Alice's and Bob's face images as $\mathcal{D}_{aux}$. PreImageGAN exploits *unlabeled* auxiliary samples to complement knowledge extracted from the model $f$.

## 1.1 OUR CONTRIBUTION

The contribution of this study is summarized as follows.

- We formulate the Classifier-to-Generator (C2G) Attack, which estimates the training sample distribution when a classification model and auxiliary samples are given(Section 3)
- We propose PreImageGAN as an algorithm for the C2G attack. The proposed method estimates the sample generation model using the interpolation ability of GANs even when the auxiliary samples used by the adversary is not drawn from the same distribution as the training sample distribution (Section 4)

---

[1]Radford et al. (2015) reported GANs could generate intermediate images between two different images. Also, Radford et al. (2015) realizes the operation of latent vectors. For example, by subtracting a latent vector of a man's face from a face image of a man wearing glasses, and then adding a latent vector of a female's face, then the GAN can generate the woman's face image wearing glasses.

- We demonstrate the performance of C2G attack with PreImageGAN using EMNIST (alphanumeric image dataset) and FaceScrub (face image dataset). Experimental results show that the adversary can estimate the sample distribution even when the adversary has no samples associated with the target label at all (Section 5)

## 2 GENERATIVE ADVERSARIAL NETWORKS

Generative Adversarial Networks (GANs) is a recently developed methodology for designing generative models proposed by Goodfellow et al. (2014). Given a set of samples, GANs is an algorithm with deep architectures that estimates the sample-generating distribution. One significant property of GANs is that it is expected to be able to accurately estimate the sample distribution even when the sample space is in the high dimensional space, and the target distribution is highly complex, such as face images or natural images. In this section, we introduce the basic concept of GANs and its variants.

The learning algorithm of GANs is formulated by minimax games consisting of two players, *generator* and *discriminator* (Goodfellow et al. (2014)). Generator $G$ generates a fake sample $G(\mathbf{z})$ using a random number $\mathbf{z} \sim d_Z$ drawn from any distribution (say, uniform distribution). Discriminator $D$ is a supervised model and is trained so that it outputs 1 if the input is a real sample $\mathbf{x} \sim d_X$ drawn from the sample generating distribution $d_X$; it outputs 0 or $-1$ if the input is a fake sample $G(\mathbf{z})$. The generator is trained so that the discriminator determines a fake sample as a real sample.

By training the generator under the setting above, we can expect that samples generated from $G(\mathbf{z})$ for arbitrary $\mathbf{z}$ are indistinguishable from real samples $\mathbf{x} \sim d_X$. Letting $Z$ be the random variable of $d_Z$, $G(Z)$ can be regarded as the distribution of samples generated by the generator. Training of GANs is known to be reduced to optimization of $G$ so that the distribution between $G(Z)$ and the data generating distribution $d_X$ is minimized in a certain type of divergence (Goodfellow et al. (2014)).

Training of GAN proposed by Goodfellow et al. (2014) (VanillaGAN) is shown to be reduced to minimization e of Jensen Shannon (JS) divergence of $G(Z)$ and $d_X$. Minimization of JS-divergence often suffers gradient explosion and mode collapse (Goodfellow et al. (2014), Arjovsky & Bottou (2017)). To overcome these problems, Wasserstein-GAN (WGAN), GAN that minimizes Wasserstein distance between $G(Z)$ and $d_X$, was proposed (Arjovsky et al. (2017), Arjovsky & Bottou (2017)). As a method to stabilize convergence behavior of WGAN, a method to add a regularization term called *Gradient Penalty (GP)* to the loss function of the discriminator was introduced (Gulrajani et al. (2017)).

Given a set of labeled samples $\{(\mathbf{x}, \mathbf{c}), \cdots\}$ where $\mathbf{c}$ denotes the label, Auxiliary Classifier GAN (ACGAN) was proposed as a GAN to estimate $d_{X|C=\mathbf{c}}$, sample distribution conditioned by label $\mathbf{c}$ (Odena et al. (2016)). Differently from VanillaGAN, the generator of ACGAN takes as input a random noise $\mathbf{z}$ and a label $\mathbf{c}$. Also, the discriminator of ACGAN is trained to predict a label of sample in addition to estimation of real or fake samples. In the learning process of ACGAN, generator is trained so that discriminator predicts correctly the label of generated sample in addition. The generator of ACGAN can generate samples with a label specified arbitrarily. For example, when $x$ corresponds to face images and $c$ corresponds to age or gender, ACGAN can generate images with specifying the age or gender (Mirza & Osindero (2014), Gauthier (2014)). In our proposed algorithm introduced in the latter sections, we employ WGAN and ACGAN as building blocks.

## 3 PROBLEM FORMULATION

### 3.1 PROBABILISTIC DISCRIMINATION MODEL

We consider a supervised learning setting. Let $\mathbb{T}$ be the label set, and $\mathbb{X} \subseteq \mathbb{R}^d$ be the sample domain where $d$ denotes the sample dimension. Let $\rho_t$ be the distribution of samples in $\mathbb{X}$ with label $t$. In face recognition, $\mathbf{x} \in \mathbb{X}$ and $t \in \mathbb{T}$ correspond to a (face) image and an individual, respectively. $\rho_t$ thus denotes the distribution of face images of individual $t$.

We suppose the images contained in the training dataset are associated with a label subset $\mathbf{T}_{\text{tr}} \subset \mathbb{T}$. Then, the training dataset is defined as $\mathcal{D}_{\text{tr}} = \{(\mathbf{x}, t) | \mathbf{x} \in \mathbb{X}, t \in \mathbf{T}_{\text{tr}}\}$. We denote the random

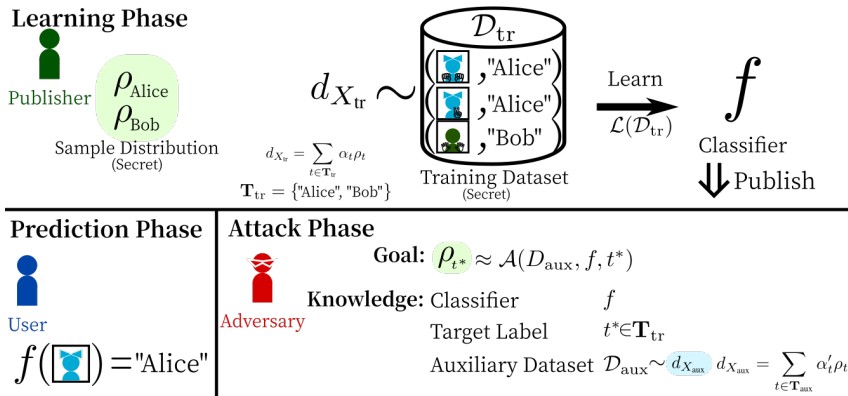

Figure 1: **Outline of Classifier-to-Generator (C2G) Attack.** The publisher trains classifier $f$ from training data $\mathcal{D}_{tr}$ and publishes $f$ to the adversary. However, the publisher does not wish to leak training data $\mathcal{D}_{tr}$ and sample generating distribution $\rho_t$ by publishing $f$. The goal of the adversary is to learn the publisher's private distribution $\rho_{t*}$ for any $t^* \in \mathbf{T}_{tr}$ specified by the adversary provided model $f$, target label $t^*$ and (unlabeled) auxiliary samples $\mathcal{D}_{aux}$.

variables associated with $(\mathbf{x}, t)$ by $(X_{tr}, T_{tr})$. Then, the distribution of $X_{tr}$ is given by a mixture distribution

$$d_{X_{tr}} = \sum_{t \in \mathbf{T}_{tr}} \alpha_t \rho_t \tag{1}$$

where $\sum_{t \in \mathbf{T}_{tr}} \alpha_t = 1, \alpha_t > 0$ for all $t \in \mathbf{T}_{tr}$. In the face recognition task example again, a training sample consists of a pair of an individual $t$ and his/her face image $\mathbf{x}$, $(\mathbf{x}, t)$ where $\mathbf{x} \sim \rho_t$.

Next, we define the probabilistic discrimination model we consider in our problem. Let $\mathbb{Y}$ be a set of $|\mathbf{T}_{tr}|$-dimension probability vector, $\Delta^{|\mathbf{T}_{tr}|}$. Given a training dataset $\mathcal{D}_{tr}$, a learning algorithm $\mathcal{L}$ gives a probabilistic discrimination model $f : \mathbb{X} \to \mathbb{Y}$ as $f = \mathcal{L}(\mathcal{D}_{tr})$. Here the $t$th element of the output $(f(\mathbf{x}))_t$ of $f$ corresponds to the probability with which $\mathbf{x}$ has label $t$. Letting $T_{tr}$ and $X_{tr}$ represents the random variable of $\mathbf{T}_{tr}$ and $d_{X_{tr}}$, $f$ is the approximation of $\Pr[T_{tr}|X_{tr}]$.

## 3.2 CLASSIFIER-TO-GENERATOR ATTACK

We define the *Classifier-to-Generator Attack (C2G Attack)* in this section. We consider two stake-holders, *publisher* and *adversary* in this attack. The publisher holds training dataset $\mathcal{D}_{tr}$ drawn from $d_{X_{tr}}$ and a learning algorithm $\mathcal{L}$. She trains model $f = \mathcal{L}(\mathcal{D}_{tr})$ and publishes $f$ to the adversary. We suppose training dataset $\mathcal{D}_{tr}$ and data generating distribution $\rho_t$ for any $t \in \mathbf{T}_{tr}$ is private or confidential information of the publisher, and the publisher does not wish to leak them by publishing $f$.

Given $f$ and $\mathbf{T}_{tr}$, the adversary aims to obtain $\rho_{t*}$ for any label $t^* \in \mathbf{T}_{tr}$ specified by the adversary. We suppose the adversary can make use of an auxiliary dataset $\mathcal{D}_{aux}$ drawn from underlying distribution $d_{X_{aux}}$ as background knowledge. $\mathcal{D}_{aux}$ is a set of samples associated with labels in $\mathbf{T}_{aux} \subset \mathbb{T}$. We remark that $\mathcal{D}_{aux}$ is defined as a set of samples associated with a specific set of labels, however, in our algorithm described in the following sections, we do not require that samples in $\mathcal{D}_{aux}$ are labeled. Then, the underlying distribution $d_{X_{aux}}$ is defined as follows:

$$d_{X_{aux}} = \sum_{t \in \mathbf{T}_{aux}} \alpha'_t \rho_t \tag{2}$$

where $\sum_{t \in \mathbf{T}_{aux}} \alpha'_t = 1, \alpha'_t > 0$ for all $t \in \mathbf{T}_{aux}$.

The richness of the background knowledge can be determined by the relation between $\mathbf{T}_{tr}$ and $\mathbf{T}_{aux}$. When $\mathbf{T}_{tr} = \mathbf{T}_{aux}$, $d_{X_{tr}} = d_{X_{aux}}$ holds. That is, the adversary can make use of samples drawn from the distribution that is exactly same as that of the publisher. In this sense, this setting is the most advantageous to the adversary. If $t^* \notin \mathbf{T}_{aux}$, the adversary cannot make use of samples with

the target label $t^*$; this setting is more advantageous to the publisher. As the overlap between $\mathbf{T}_{\mathrm{tr}}$ and $\mathbf{T}_{\mathrm{aux}}$ increases, the situation becomes more advantageous to the adversary. Discussions on the background knowledge of the adversary are given in 3.4 in detail.

The goal of the adversary is to learn the publisher's private distribution $\rho_{t^*}$ for any $t^* \in \mathbf{T}_{\mathrm{tr}}$ specified by the adversary provided model $f$, target label $t^*$ and auxiliary (unlabeled) samples $\mathcal{D}_{\mathrm{aux}}$. Let $\mathcal{A}$ be the adversary's attack algorithm. Then, the attack by the adversary can be formulated by

$$\hat{\mathbf{x}}^{(t^*)} \sim \mathcal{A}(f, \mathcal{D}_{\mathrm{aux}}, t^*)$$

where the output of $\mathcal{A}$ is a distribution over $\mathbb{X}$. In the face recognition example of Alice and Bob again, when the target label of the adversary is $t^* =$Alice, the objective of the adversary is to estimate the distribution of face images of Alice by $\mathcal{A}(f, \mathcal{D}_{\mathrm{aux}}, t^*)$.

### 3.3 Evaluation of the C2G Attack

The objective of the C2G attack to estimate $\rho_{t^*}$, the private data generating distribution of the publisher. In principle, the measure of the success of the C2G attack is evaluated with the quasi-distance between the underlying distribution $\rho_{t^*}$ and the estimated generative model $\mathcal{A}(f, \mathcal{D}_{\mathrm{aux}}, t^*)$. If the two distributions are close, we can confirm that the adversary successfully estimates $\rho_{t^*}$. However, $\rho_{t^*}$ is unknown, and we cannot evaluate this quasi-distance directly.

Instead of evaluating the distance of the two distributions directly, we evaluate the attack algorithm empirically. We first prepare a classifier $f'$ that is trained with $\mathcal{D}_{\mathrm{tr}}$ using a learning algorithm different from $f$. We then give samples drawn from $\mathcal{A}(f, \mathcal{D}_{\mathrm{aux}}, t^*)$ to $f'$ and evaluate the probability of which the label of the given samples are predicted as $t^*$. We expect that the classifier $f'$ would label samples drawn from $\mathcal{A}(f, \mathcal{D}_{\mathrm{aux}}, t^*)$ as $t^*$ with high probability if $\mathcal{A}(f, \mathcal{D}_{\mathrm{aux}}, t^*)$ successfully estimates $\rho_{t^*}$. Considering the possibility that $\mathcal{A}(f, \mathcal{D}_{\mathrm{aux}}, t^*)$ overfits to $f$, we employ another classifier $f'$ for this evaluation. This evaluation criterion is the same as the *inception accuracy* introduced for ACGAN by Odena et al. (2016). In our setting, since our objective is to estimate the distribution concerning a specific label $t^*$, we employ the following inception accuracy:

$$\mathrm{Pr}_{\mathbf{x} \sim \mathcal{A}(f, \mathcal{D}_{\mathrm{aux}}, t^*)} \left[ \arg\max_{t \in \mathbf{T}_{\mathrm{tr}}} (f'(x))_t = t^* \right]$$

We remark that the generated model with a high inception accuracy is not always a reasonable estimation of $\rho_{t^*}$. Discrimination models with a deep architecture are often fooled with artifacts. For example, Nguyen et al. (2014) reported that images look like white noise for humans can be classified as a specific object with high probability. For this reason, we cannot conclude that a model with a high inception accuracy always generates meaningful images. To avoid this, the quality of generated images should be subjectively checked by humans.

The evaluation criterion of the C2G Attack we employed for this study is similar to those for GANs. Since the objective of GANs and the C2G attack is to estimate unknown generative models, we cannot employ the pseudo distance between the underlying generating distribution and the estimated distribution. The evaluation criterion of GANs is still an open problem, and subjective evaluation is needed for evaluation of GANs (Goodfellow (2017)). In this study, we employ both the inception accuracy and subjective evaluation for performance evaluation of the C2G attack.

### 3.4 Background Knowledge of Adversary

The richness of the background knowledge of the adversary affects the performance of the C2G attack significantly. We consider the following three levels of the background knowledge. In the following, let $\mathbf{T}_{\mathrm{aux}}$ be the set of labels of samples generated by the underlying distribution of the auxiliary data, $d_{X_{\mathrm{aux}}}$. Also, let $\mathbf{T}_{\mathrm{tr}}$ be the set of labels of samples generated by the underlying distribution of the training data.

- **Exact same:** $\mathbf{T}_{\mathrm{tr}} = \mathbf{T}_{\mathrm{aux}}$
  In this setting, we suppose $\mathbf{T}_{\mathrm{tr}}$ is exactly same as the $\mathbf{T}_{\mathrm{aux}}$. Since $\mathcal{D}_{\mathrm{aux}}$ follows $d_{X_{\mathrm{tr}}}$, $\mathcal{D}_{\mathrm{aux}}$ contains samples with the target label. That is, the adversary can obtain samples labeled with the target label. The background knowledge of the adversary in this setting is the most powerful among the three settings.

- **Partly same:**$t^* \notin \mathbf{T}_{\text{aux}}, \mathbf{T}_{\text{aux}} \subset \mathbf{T}_{\text{tr}}$
  In this setting, $\mathbf{T}_{\text{aux}}$ and $\mathbf{T}_{\text{tr}}$ are overlapping. However, $\mathbf{T}_{\text{aux}}$ does not contain the target label. That is, the adversary cannot obtain samples labeled with the target label. In this sense, the background knowledge of the adversary in this setting is not as precise as that in the former setting.

- **Mutually exclusive:** $\mathbf{T}_{\text{aux}} \cap \mathbf{T}_{\text{tr}} = \emptyset$
  In this setting, we suppose $\mathbf{T}_{\text{aux}}$ and $\mathbf{T}_{\text{tr}}$ are mutually exclusive, and the adversary cannot obtain samples labeled with the target label. In this setting, the adversary cannot obtain any samples with labels used for training of model $f$. In this sense, the background knowledge of the adversary in this setting is the poorest among the three settings.

## 4 PreImageGAN

### 4.1 Outline of PreImageGAN

In this section, we propose PreImageGAN that works as an algorithm that achieves C2G attack. Given the model $f \simeq d_{Y_{\text{tr}}|X_{\text{tr}}}$, the goal of PreImageGAN is to estimate $\left(d_{X_{\text{tr}}|Y_{\text{tr}}=\mathbf{y}^{(t^*)}}\right) = \rho_{t^*}$ for target label $t^*$ specified by the adversary where $\mathbf{y}^{(t^*)}$ is the one-hot vector in which the element corresponds to $t^*$ is activated.

In the adversary can draw samples from $d_{X_{\text{tr}}|Y_{\text{tr}}=\mathbf{y}^{(t^*)}}$, this generative model can be straightforwardly estimated with existing GANs. However, in the setting of the C2G attack, what the adversary can utilize for the C2G attack is the probabilistic classification model $f$ that is expected to be similar to $d_{Y_{\text{tr}}|X_{\text{tr}}}$ and auxiliary samples that are drawn from auxiliary distribution $d_{X_{\text{aux}}}$ only. So the target distribution cannot be estimated in a straightforward manner. PreImageGAN first gives labels to auxiliary samples with the given model as $\mathbf{y} = f(\mathbf{x})$ and estimates $d_{X_{\text{aux}}|Y_{\text{aux}}}$ using $\{(\mathbf{x}, \mathbf{y})|\mathbf{x} \in \mathcal{D}_{\text{aux}}\}$ by optimizing the objective function defined below.

If the sample distribution of the auxiliary samples is close to that of the true underlying distribution, we can expect that estimation of $d_{X_{\text{aux}}|Y_{\text{aux}}}$ can be used as an approximation of $d_{X_{\text{tr}}|Y_{\text{tr}}}$. More specifically, we can obtain the generative model of the target label $d_{X_{\text{aux}}|Y_{\text{aux}}=\mathbf{y}^{(t^*)}}$ by specifying the one-hot vector of the target label as the condition.

As we mentioned in Section 3.4, the sample generating distribution of the auxiliary samples is not necessarily equal or close to the true sample generating distribution. In the "partly same" setting or "mutually exclusive" setting, $\mathcal{D}_{\text{aux}}$ does not contain samples labeled with $t^*$ at all. It is well known that GANs can generate samples with interpolating latent variables (Berthelot et al. (2017), Radford et al. (2015)). We expect that PreImageGAN generates samples with the target label by interpolation of latent variables of given samples without having samples of the target label. More specifically, if latent variables of given auxiliary samples are diverse enough and $d_{X_{\text{aux}}|Y_{\text{aux}}}$ well approximates the true sample generating distribution, we expect that GAN can generate samples with the target label by interpolating obtained latent variables of auxiliary samples without having samples with the target label.

Figure 2 describes a conceptual illustration of generation of image "9" when PreImageGAN takes images of alphabets (A-Z,a-z) as auxiliary samples and $f(\mathbf{x}) = \Pr[\text{number}|\mathbf{x}]$ as a probabilistic classification model. The auxiliary samples do not contain images of numbers whereas PreImageGAN generates images close to "9" by interpolating latent variables of alphabets close to "9", such as "g" or "Q".

### 4.2 Objective Function

In PreImageGAN, a conditional attribute $\mathbf{y} \sim d_{Y_{\text{aux}}}$ in addition to noise $\mathbf{z} \sim d_Z$ is used to generate samples. Similarly to other GANs, $d_Z$ can be arbitrarily determined by the learner (adversary). Given $f$ and $\mathcal{D}_{\text{aux}}$, $d_{Y_{\text{aux}}}$ can be empirically estimated from the set of predictions $\{f(\mathbf{x})|\mathbf{x} \in \mathcal{D}_{\text{aux}}\}$.

Generator $G : (\mathbb{Z}, \mathbb{Y}) \to \mathbb{X}$ of PreImageGAN generates fake samples $\mathbf{x}_{\text{fake}} = G(\mathbf{z}, \mathbf{y})$ using random draws of $\mathbf{y}$ and $\mathbf{z}$. After the learning process is completed, we expect generated fake samples $\mathbf{x}_{\text{fake}}$ satisfy $f(\mathbf{x}_{\text{fake}}) = \mathbf{y}$. On the other hand, discriminator $D : \mathbb{X} \to \mathbb{R}$ takes as input a sample $\mathbf{x}$ and discriminates whether it is a generated fake sample $\mathbf{x}_{\text{fake}}$ or a real sample $\mathbf{x}_{\text{real}} \in \mathcal{D}_{\text{aux}}$.

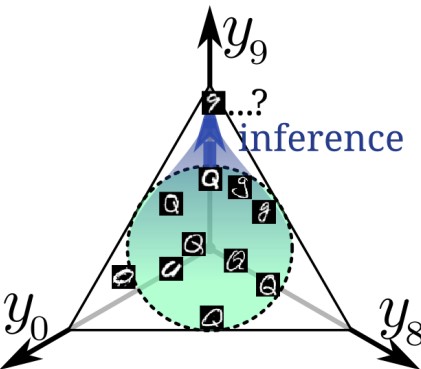

Figure 2: Inference on the space of **y**. Here, we suppose the adversary has auxiliary samples labeled with alphabets only and a probabilistic classification model that takes an image of a number and outputs the corresponding number. The axis corresponds to an element of the probabilistic vector outputted by the classification model. For example, $y_9 = (f(\mathbf{x}))_9$ denotes the probability that the model discriminates the input image as "9". The green region in the figure describes the spanned by auxiliary samples in $\mathcal{D}_{\text{aux}}$. $\mathcal{D}_{\text{aux}}$ does not contain images of numbers classified so that $y_9 = 1$ or $y_8 = 1$ whereas PreImageGAN generates samples close to "9" by interpolating latent variables of images that are close to "9" such as "Q" and "g".

With these requirements, the objective function of $G$ and $D$ is formulated as follows

$$\min_{G} \max_{\|D\|_{\text{L}} \leq 1} \mathrm{E}_{\mathbf{x} \sim d_X} [D(\mathbf{x})]$$
$$- \mathrm{E}_{\mathbf{z} \sim d_Z, \mathbf{y} \sim d_{Y_{\text{aux}}}} [D(G(\mathbf{z}, \mathbf{y}))] \qquad (3)$$
$$- \gamma \mathrm{E}_{\mathbf{z} \sim d_Z, \mathbf{y} \sim d_{Y_{\text{aux}}}} [\text{similarity}(f(G(\mathbf{z}, \mathbf{y})), \mathbf{y})]$$

where $\| \cdot \|_{\text{L}} \leq 1$ denotes $\alpha$-Lipschitz functions with $\alpha \leq 1$.

By maximizing the first and second term concerning $D$, Wasserstein distance between the marginal of the generator $\int G(Z, Y_{\text{aux}}) dY_{\text{aux}}$ and the generative distribution of auxiliary samples $d_{X_{\text{aux}}}$ is minimized. By maximizing the similarity between **y** and $f(G(\mathbf{z}, \mathbf{y}))$, $G$ is trained so that samples generated from $G(\mathbf{z}, \mathbf{y})$ satisfy $f(G(\mathbf{z}, \mathbf{y})) = \mathbf{y}$. $\gamma \geq 0$ works as a parameter adjusts the effect of this term.

For sample generation with PreImageGAN, $G(Z, \mathbf{y}^{(t^*)})$ is utilized as the estimation of $\rho_{t^*}$. We here remark that model $f$ is regarded as a constant in the learning process of GAN and used as it is.

## 5 EXPERIMENTS

In this section, we show that the proposed method enables to perform the C2G attack with experiments. We experimentally demonstrate that the adversary can successfully estimate $\rho_{t^*}$ given classifier $f$ and the set of unlabeled auxiliary samples $\mathcal{D}_{\text{aux}} = \{\mathbf{x} | \mathbf{x} \in \mathbb{X}\}$ even in the partly same setting and the mutually exclusive setting under some conditions.

### 5.1 EXPERIMENTAL SETUP

For demonstration, we consider a hand-written character classification problem (EMNIST) and a face recognition problem (FaceScrub). We used the Adam optimizer ($\alpha = 2 \times 10^{-4}$, $\beta_1 = 0.5$, $\beta_2 = 0.999$) for the training of the generator and discriminator. The batch size was set as 64. We set the number of discriminator (critic) iterations per each generator iteration $n_{\text{cric}} = 5$. To enforce the 1-Lipschitz continuity of the discriminator, we add a gradient penalty (GP) term to the loss function of the discriminator (Gulrajani et al. (2017)) and set the strength parameter of GP as $\lambda = 10$. We used 128-dim uniform random distribution $[-1, 1]^{128}$ as $d_Z$. We estimated $d_{Y_{\text{aux}}}$ empirically from $\{f(\mathbf{x}) | \mathbf{x} \in \mathcal{D}_{\text{aux}}\}$ using kernel density estimation where the bandwidth is 0.01, and the Gaussian kernel was employed.

## 5.2 EMNIST: HAND-WRITTEN CHARACTERS CLASSIFIER

EMNIST consists of grayscale 28x28 pixel images from 62 alphanumeric characters (0-9A-Za-z). We evaluate the C2G attack with changing the richness of the adversary's background knowledge as discussed in Section 3.4 (exact same, partly same, and mutually exclusive) to investigate how the richness of the auxiliary data affects the results.

Also, to investigate how the choice of the auxiliary data affects the results, we tested two different types of target labels as summarized in Table 1 (lower-case target) and Table 2 (numeric target). In the former setting, an alphanumeric classification model is given to the adversary. In the latter setting, a numeric classification model is given to the adversary.

### 5.2.1 C2G ATTACK TARGETING LOWER-CASE CHARACTERS

In this setting, the target label $t^*$ was set as lower-case characters ($t^* \in \{a, b, \dots, z\}$) (Table 1). In the exact/partly same setting, an alphanumeric classifier (62 labels) is given to the adversary where the classifier is trained for ten epochs and achieved test accuracy 0.8443. In the mutually exclusive setting, an alphanumeric classifier (36 labels) given to the adversary where the classifier is trained for ten epochs and achieved test accuracy 0.9202. See Table 1 for the detailed settings. In the training process of PreImageGAN, we trained the generator for 20k iterations. We set the initial value of $\gamma$ to 0, incremented gamma by 0.001 per generator iteration while $\gamma$ is kept less than 10.

Fig. 3 represents the results of the C2G attack with targeting lower-case characters against given alphanumeric classification models. Alphabets whose lower-case and upper-case letters are similar (e.g., C, K) are easy to attack. So, we selected alphabets whose lower-case letter and upper-case letter shapes are dissimilar in Fig. 3.

In the exact same setting, we can confirm that the PreImageGAN works quite successfully. In the partly same setting, some generated images are disfigured compared to the exact same setting (especially when $t^* = q$) while most of the target labels are successfully reconstructed. In the mutually exclusive setting, some samples are disfigured (especially when $t^* = h, i, q$) while remaining targets are successfully reconstructed. As an extreme case, we tested the case when the auxiliary data consists of images drawn from uniform random, and we observed that the C2G attack could generate no meaningful images (See Fig. 7 in Appendix A). From these results, we can conclude that the C2G attack against alphanumeric classifiers works successfully except several although there is an exception in the mutually exclusive setting.

### 5.2.2 C2G ATTACK TARGETING NUMERIC CHARACTERS

We also tested the C2G attack when the target label $t^*$ was set as numeric characters ($t^* \in \{0, 1, \dots, 9\}$). In the exact/partly same setting, an alphanumeric classifier (62 labels, test accuracy 0.8443.) is given to the adversary. In the mutually exclusive setting, a numeric classifier (10 labels, test accuracy 0.9911) given to the adversary where the classifier is trained for ten epochs and achieved test accuracy 0.9202. See Table 2 for the detailed settings. PreImageGAN was trained in the same setting as the previous subsection.

Fig. 4 represents the results of the C2G attack with targeting numeric characters against given classification models. In the exact/partly same setting, the PreImageGAN works quite successfully as well with some exceptions; e.g., "3" and "7" are slightly disfigured in the partly same setting. On the other hand, in the mutually exclusive setting, images targeting "0" and "1" look like the target numeric characters while remaining images are disfigured or look like other alphabets. As shown from these results, in the mutually exclusive setting, the C2G attack against alphabets works well with while it fails when targeting numeric characters.

One of the reasons for this failure is in the incompleteness of the classifier given to the attacker. More precisely, when the classifier recognizes images with non-target labels as a target labels falsely, C2G attack fails. For example, in Fig 4, images of "T" are generated as images of "7" in the mutually exclusive setting. This is because the given classifier recognizes images of "T" as "7" falsely, and the PreImageGAN thus generates images like "T" as "7". See Table 8 in Appendix B; many alphabets are recognized as numeric characters falsely. As long as the given classification model recognizes

Table 1: Summary of the settings in the C2G attack against an alphanumeric classifiadtion model (EMNIST)

| | Training samples $\mathcal{D}_{tr}$ and classifier $f$ | Auxiliary samples $\mathcal{D}_{aux}$ | Adversary's target $t^*$ |
|---|---|---|---|
| Exact same | upper/lowercase, numeric | upper/lowercase, numeric | lowercase |
| Partly same | upper/lowercase, numeric | uppercase | lowercase |
| Mutually exclusive | lowercase, numeric | uppercase | lowercase |

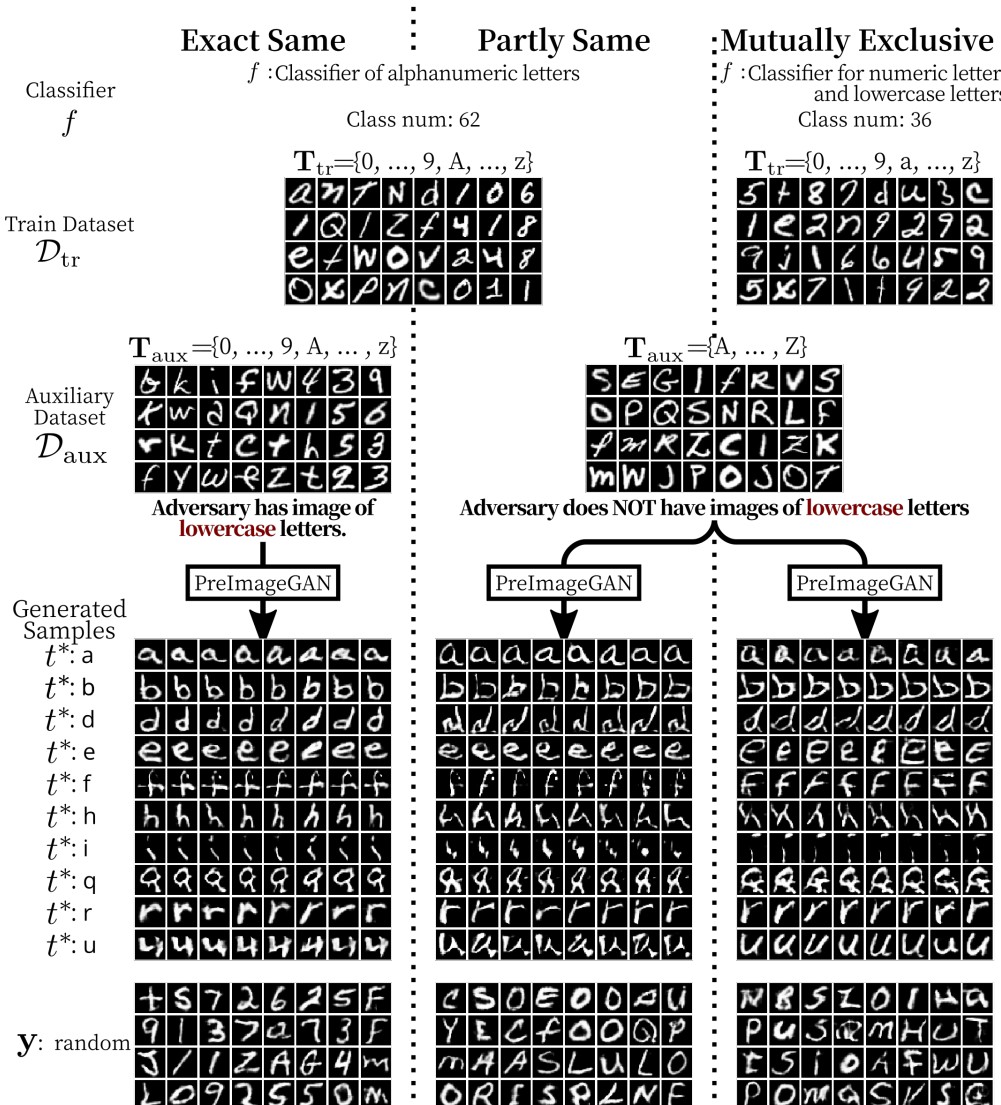

Figure 3: C2G attack against an alphanumeric classifier with changing the richness of the background knowledge of the adversary targeting lowercase letters. The samples in the bottom row ("**y**: random") are generated when **y** is randomly drawn from empirically estimated $d_{Y_{aux}}$.

non-target characters as target characters, the C2G attack cannot generate images of the target labels correctly.

In Fig 3, images of "h", "i", and "q" are disfigured in the mutually exclusive setting. This disfiguring occurs for a different reason (see Table 6 in Appendix B; no significant false recognition can be found for these characters). We consider this is because the image manifold that the classifier recognizes as the target character does not exactly fit the image manifold of the target character. Since the images generated by the C2G attack for "h," "i," and "q" are recognized as "h," "i," and "q" by the classifier with a very high probability, the images work as adversarial examples. Detailed analysis of the failure of the C2G attack would give us a hint to establish defense methods against the C2G attack. Detailed consideration of this failure remains as future work.

Finally, we evaluate the quality of the C2G attack; we measured the inception accuracy using. Here we employed a ResNet-based network architecture as $f'$ (see Table E in detail). As a baseline, we trained ACGAN [Odena et al. (2016)] with the same training samples $\mathcal{D}_{\text{tr}}$ and evaluated the inception accuracy. In the exact same setting, the inception accuracy is almost equal to ACGAN. From the results, we can see that the inception scores drop as the background knowledge of the adversary becomes poorer. This result indicates that the background knowledge of the adversary affects the result of the C2G attack significantly.

## 5.3 FACESCRUB: FACE RECOGNITION

FaceScrub dataset consists of color face images (530 persons). We resized images to 64x64 pixel images for experiments and evaluated the C2G attack in the mutually exclusive setting. In detail, we picked up 100 people as $\mathbf{T}_{\text{tr}}$ (see Appendix D for the list) and used remaining 430 persons as $\mathbf{T}_{\text{aux}}$ (mutually exclusive setting). If the adversary can generate face images of the 100 people in $\mathbf{T}_{\text{tr}}$ by utilizing model $f$ recognizing $\mathbf{T}_{\text{tr}}$ and face images with labels in $\mathbf{T}_{\text{aux}}$, we can confirm that the C2G attack works successfully.

$\mathcal{D}_{\text{tr}}$ consists of 12k images, and $\mathcal{D}_{\text{aux}}$ consists of 53k images. $f$ is trained on $\mathcal{D}_{\text{tr}}$ for ten epochs and achieved test accuracy 0.8395. In training PreImageGAN, we train the generator for 130k iterations. We set the initial value of $\gamma$ to 0, incremented gamma by 0.0001 per generator iteration while $\gamma$ is kept less than 10.

Fig. 5 represents the results of the C2G attack against the face recognition model. Those samples are randomly generated without human selection. The generated face images well capture the features of the face images in the training samples. From the results, we can see that the C2G attack successfully reconstruct training samples from the face recognition model without having training samples in the mutually exclusive setting.

As byproducts, we can learn what kind of features the model used for face recognition with the results of the C2G attack. For example, all generated face images of Keanu Reeves wear the mustache. This result implies that $f$ exploits his mustache to recognize Keanu Reeves.

One may concern that the PreImageGAN simply picks up images in the auxiliary samples that look like the target quite well, but labeled as some other person. To show that the PreImageGAN does not simply pick up similar images to the target, but it generates images of the target by exploiting the features of face images extracted from the auxiliary images, we conducted two experiments.

First, we evaluated the probability with which classifier $f$ recognizes images in the auxiliary dataset and images generated by the PreImageGAN as the target (Keanu Reeves and Marg Helgenberger) in Figure 6. The probabilities are sorted, and the top 500 results are shown in the figure.

The orange lines denote the probability with which the images in the auxiliary dataset is recognized as the target. A few images have a high probability ($>0.80$), but much less than the probabilities with which the target image in training data(blue lines) is recognized as the target ($>0.80$, 80 images). This indicates that the auxiliary samples do not contain images that are quite similar to the targets. The green lines denote the probability with which the images generated by the PreImageGAN are recognized as the target. As seen from the results, the generated images are recognized as the target with extremely high probability ($> 0.95$). This suggests that the PreImageGAN could generate images recognized as a target with high probability from samples not recognized as the target.

Table 2: Summary of the settings of C2G attack against a numeric classification model (EMNIST)

| | Training samples $\mathcal{D}_{tr}$ and classifier $f$ | Auxiliary samples $\mathcal{D}_{aux}$ | Adversary's target $t^*$ |
|---|---|---|---|
| Exact same | upper/lowercase, numeric | numeric | numeric |
| Partly same | upper/lowercase, numeric | upper/lowercase | numeric |
| Mutually exclusive | numeric | upper/lowercase | numeric |

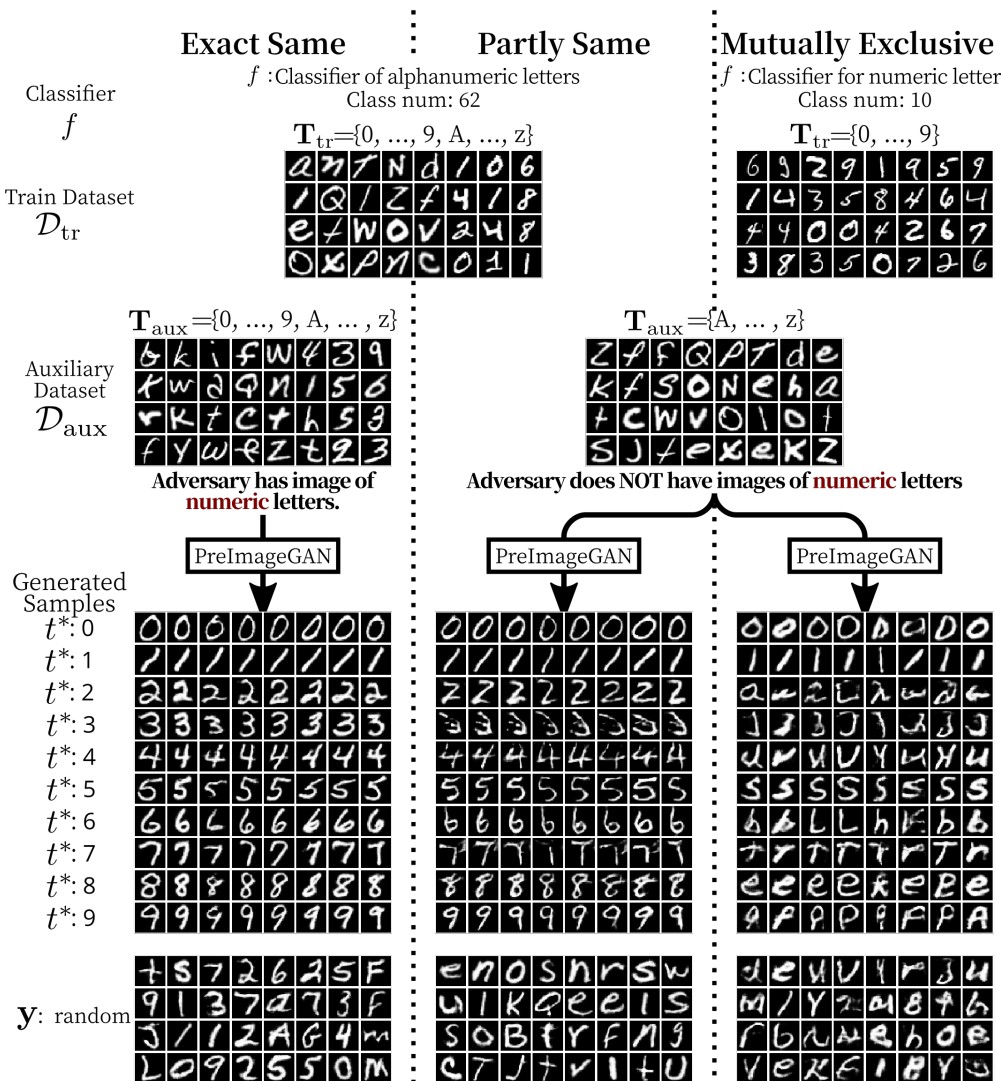

Figure 4: C2G attack against a numeric classification model with changing the richness of the background knowledge of the adversary targeting numeric letters. The samples in the bottom row ("**y**: random") are images generated when **y** is randomly drawn from empirically estimated $d_{Y_{aux}}$.

Table 3: Inception Accuracy with changing background knowledge of the adversary (C2G attack against alphanumeric classifier)

| Setting | target label: $t^*$ | | | | | | | | | |
|---|---|---|---|---|---|---|---|---|---|---|
| | a | b | d | e | f | h | i | q | r | u |
| Baseline (ACGAN) | 1.00 | 1.00 | 1.00 | 1.00 | 1.00 | 1.00 | 1.00 | 1.00 | 1.00 | 1.00 |
| Exact same | 1.00 | 0.99 | 1.00 | 0.99 | 0.00 | 1.00 | 1.00 | 0.36 | 0.99 | 0.58 |
| Partly same | 0.99 | 1.00 | 0.89 | 0.67 | 0.00 | 0.71 | 0.08 | 0.33 | 0.99 | 0.60 |
| Mutually exclusive | 0.10 | 0.81 | 0.76 | 0.95 | 1.00 | 0.86 | 0.94 | 0.84 | 0.99 | 1.00 |

Table 4: Inception Accuracy with changing background knowledge of the adversary (C2G attack against numeric classifier)

| Setting | target label: $t^*$ | | | | | | | | | |
|---|---|---|---|---|---|---|---|---|---|---|
| | 0 | 1 | 2 | 3 | 4 | 5 | 6 | 7 | 8 | 9 |
| Baseline (ACGAN) | 1.00 | 1.00 | 1.00 | 1.00 | 1.00 | 1.00 | 1.00 | 1.00 | 1.00 | 1.00 |
| Exact same | 1.00 | 1.00 | 1.00 | 1.00 | 1.00 | 1.00 | 0.99 | 1.00 | 1.00 | 1.00 |
| Partly same | 1.00 | 1.00 | 0.89 | 0.71 | 0.99 | 1.00 | 0.90 | 0.53 | 0.79 | 0.99 |
| Mutually exclusive | 0.97 | 1.00 | 0.59 | 0.39 | 0.86 | 0.98 | 0.97 | 0.61 | 0.72 | 0.48 |

Second, to demonstrate that the PreImgeGAN can generate a wide variety of images of the target, we generated images that interpolate two targets (see Appendix C for the settings and the results). As seen from the results, the PreImageGAN can generate face images by exploiting both classifier $f$ and the features extracted from the auxiliary dataset.

Finally, we conducted our experiments Intel(R) Xeon(R) CPU E5-2623 v3 and a single GTX TITAN X (Maxwell), and it spends about 35 hours to complete the entire training process for the FaceScrub experiment. The computational capability of our environment is almost the same as the p2 instance (p2.xlarge) of Amazon Web Service. Usage of a p2.xlarge instance for 50 hours costs about $45. This means that the C2G attack is a quite practical attack anyone can try with a regular computational resource at low cost.

## 6 RELATED WORK

Fredrikson et al. (2014) proposed the model inversion attack against machine learning algorithms that extract private input attributes from published predicted values. Through a case study of personalized adjustment of Warfaline dosage, Fredrikson et al. (2014) showed that publishing predicted dosage amount can cause leakage of private input attributes (e.g., personal genetic information) in generalized linear regression. Fredrikson et al. (2015) presented a model inversion attack that reconstructs face images from a face recognition model. The significant difference between the C2G attack and the model inversion attack is the goal of the adversary. In the model inversion attack, the adversary tries to estimate a private input (or input attributes) $x$ from predicted values $y = f(x)$ using the predictor $f$. Thus, the adversary's goal is the private input $x$ itself in the model inversion attack. By contrast, in the C2G attack, the adversary's goal is to obtain the training sample distribution. Another difference is that the target network model. The target model of Fredrikson et al. (2015) was a shallow neural network model while ours is deep neural networks. As the network architecture becomes deeper, it becomes more difficult to extract information about the input because the output of the model tends to be more abstract (Hitaj et al. (2017)).

Hitaj et al. (2017) discussed leakage of training samples in collaborative learning based on the model inversion attack using the IcGAN (Perarnau et al. (2016)). In their setting, the adversary's goal is not to estimate training sample distribution but to extract training samples. Also, their demonstration is limited to small-scale datasets, such as MNIST dataset (hand-written digit grayscale images, 10 labels) and AT&T dataset (400 face grayscale images with 40 labels). By contrast, our experiments are demonstrated with larger datasets, such as EMNIST dataset (62 labels) and FaceScrub dataset (530 labels, 100,000+ color images).

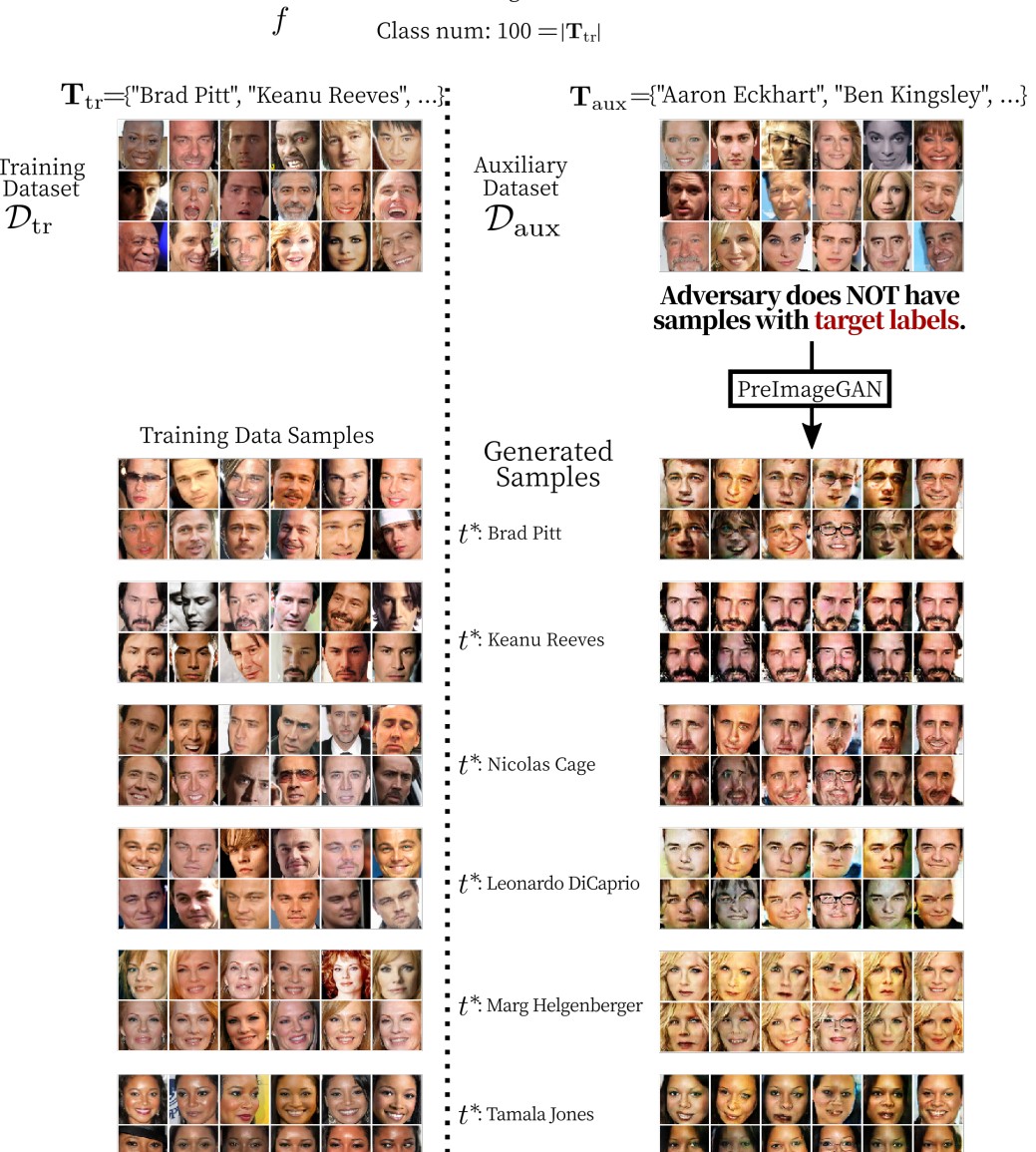

Figure 5: **FaceScrub: The results of the C2G attack against face recognition in the mutually exclusive setting.** We trained a face recognition model of 100 people (including Brad Pitt, Keanu Reeves, Nicolas Cage and Marg Helgenberger), and evaluated the C2G attack where the classification model for the 100 people is given to the adversary while no face images of the 100 people are not given. Generated samples are randomly selected, and we did not cherry-pick "good" samples. We can recognize the generated face images as the target label. This indicates that the C2G attack works successfully for the face recognition model.

Hayes et al. (2017) discussed the membership inference attack against a generative model trained by BEGAN (Berthelot et al. (2017)) or DCGAN (Radford et al. (2015)). In the membership inference attack, the adversary's goal is to determine whether the sample is contained in the private training dataset; the problem and the goal are apparently different from ours.

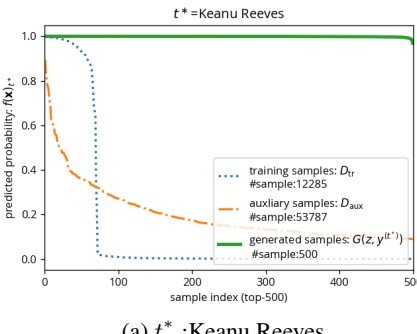 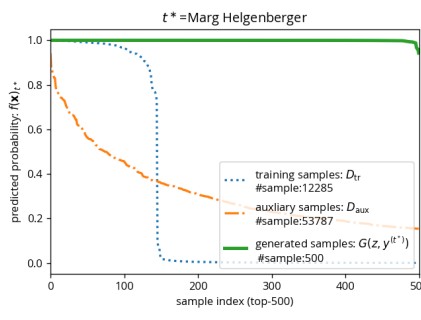

(a) $t^*$ :Keanu Reeves         (b) $t^*$ :Marg Helgenberger

Figure 6: Orange lines: probabilities with which the images in the auxiliary dataset is recognized as the target. Blue lines: probabilities with which the target image in training data is recognized as the target. Green line: probability with which the images generated by the PreImageGAN are recognized as the target. The probabilities are sorted, and the top 500 results are shown in the figure. Images in the auxiliary data set are not recognized as the target with high probability while images generated by the PreImageGAN are recognized as the target with very high probability.

Song et al. (2017) discussed malicious regularizer to memorize private training dataset when the adversary can specify the learning algorithm and obtain the classifier trained on the private training data. Their experiments showed that the adversary can estimate training data samples from the classifier when the classifier is trained with malicious regularizer. Since our setting does not assume that the adversary can specify the learning algorithm, the problem setting is apparently different from ours.

Mahendran & Vedaldi (2015) and Mahendran & Vedaldi (2016) consider the understanding representation of deep neural networks through reconstruction of input images from intermediate features. Their studies are related to ours in the sense that the algorithm exploits intermediate features to attain the goal. To the best of our knowledge, no attack algorithm has been presented to estimate private training sample distribution as the C2G attack achieves.

## 7 CONCLUSION

As described in this paper, we formulated the Classifier-to-Generator (C2G) Attack, which estimates the training sample distribution $\rho_{t*}$ from given classification model $f$ and auxiliary dataset $\mathcal{D}_{\text{tr}}$. As an algorithm for C2G attack, we proposed PreImageGAN which is based on ACGAN and WGAN. The proposed method can estimate the sample generation model using the interpolation ability of GANs even if the adversary does not have samples with training data labels. In experiments, we demonstrated the performance of C2G attack against handwritten character classifier and face recognition model. Experimental results show that the adversary can estimate the training sample distribution $\rho_{t*}$ even when the adversary does not have samples with training data labels.

### ACKNOWLEDGMENTS

To be written.

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

## A   C2G ATTACK WITH USING MEANINGLESS NOISE IMAGES AS AUXILIARY DATASET

Fig. 7 represents the results of the C2G attack when the auxiliary data consists of noisy images which are drawn from the uniform distribution. All generated images look like noise images, not numeric letters. This result reveals that the C2G attack fails when the auxiliary dataset is not sufficiently informative. More specifically, we can consider the C2G attack fails when the attacker does not have appropriate background knowledge of the training data distribution.

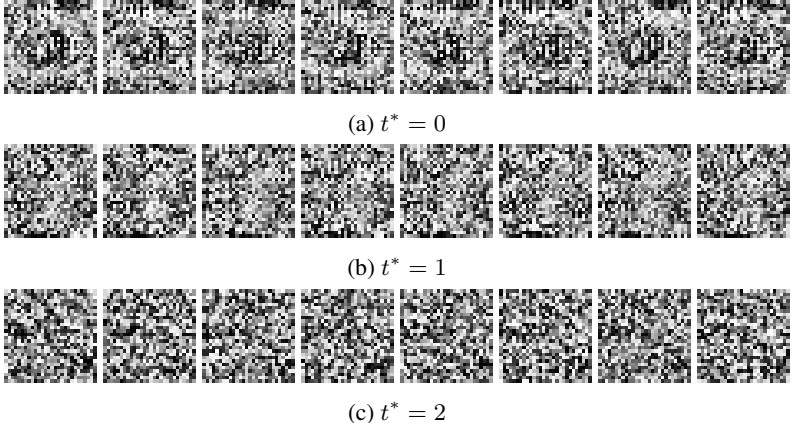

(a) $t^* = 0$

(b) $t^* = 1$

(c) $t^* = 2$

Figure 7: Images generated by the C2G attack when the target label is set as $t^* = 0, 1, 2$ and uniformly generated noise images are used as the auxiliary dataset. We used an alphanumeric letter classifier (label num:62) described in Sec. 5.2 as $f$ for this experiment.

## B    Top-5 Misclassified Characters in Alphanumeric/Numeric Classifier

Images generated by the C2G attack is significantly affected by the property of the classification model $f$ given to the C2G attack. To investigate this, we measured how often non-target characters are falsely recognized as a target character by classification model $f$ with high probability (greater than 0.9). The tables shown below in this subsection contain at most the top-five falsely-recognized characters for each target label. If no more than five characters are falsely recognized with high probability, the fields remain blank.

### B.1    False recognition of lowercase labels

We consider an alphanumeric classifier $f$ trained in the exactly/partly same setting of Table 1; table 5 represents the characters falsely recognized as the target label with high probability. Similarly, for an alphanumeric classifier $f$ trained in the mutually exclusive setting of Table 1, table 6 represents the characters falsely recognized as the target label with high probability.

Table 5: Top-five non-target characters falsely recognized as target characters (lower-case) with high probability by alphanumeric (upper/lower-case, numeric) classifier. ″(X: 0.123)″ means that the classifier misclassified 12.3%(0.123) of images of ″X″ as the target character with high probability (>0.9).

| misclassified as | Top-5 falsely recognized characters | | | | |
|---|---|---|---|---|---|
| a | (9: 0.002) | | | | |
| b | (g: 0.016) | (6: 0.002) | | | |
| d | | | | | |
| e | (c: 0.025) | (C: 0.011) | (8: 0.002) | | |
| f | | | | | |
| h | | | | | |
| i | | | | | |
| q | | | | | |
| r | (Y: 0.010) | (T: 0.006) | | | |
| u | | | | | |

Table 6: Top-five non-target characters falsely recognized as target characters (lower-case) with high probability by alphanumeric (lower-case, numeric) classifier. ″(X: 0.123)″ means that the classifier misclassified 12.3%(0.123) of images of ″X″ as the target character with high probability (>0.9).

| misclassified as | Top-5 falsely recognized characters | | | | |
|---|---|---|---|---|---|
| a | (Q: 0.106) | (u: 0.020) | (A: 0.009) | | |
| b | (g: 0.016) | | | | |
| d | (J: 0.013) | | | | |
| e | (E: 0.186) | (c: 0.050) | (C: 0.034) | (R: 0.028) | (p: 0.022) |
| f | (F: 0.244) | (E: 0.010) | | | |
| h | (m: 0.025) | (n: 0.005) | | | |
| i | | | | | |
| q | | | | | |
| r | (T: 0.006) | (C: 0.006) | | | |
| u | (U: 0.066) | | | | |

As confirmed by the results of Table 5, few letters are falsely recognized by alphanumeric classifier $f$ in the exactly/partly same setting. This results support the fact that the C2G attack works quite successfully in this setting (see Figure 3). In Table 6, ″E″ and ″F″ are falsely recognized as ″e″ and ″f″, respectively frequently, while the C2G attack could successfully generate ″e″ and ″f″ in Figure 3. This is because (″e″, ″E″) and (″f″, ″F″) have similar shapes and this false recognition of the model does not (fortunately) disfiguring in generation of the images. In Figure 3, images of ″i″ and ″q″ in the mutually exclusive setting are somewhat disfigured while no false recognition of these characters is found in Table 6. This disfiguring is supposed to occur because the classification model $f$ does not necessarily exploit the entire structure of ″i″ and ″q″; the image manifold that $f$ recognizes as ″i″ and ″q″ does not exactly fits the image manifold of ″i″ and ″q″.

## B.2 FALSE RECOGNITION OF NUMERIC LABELS

Next, we consider an alphanumeric classifier $f$ trained in the exactly/partly same setting of Table 2; table 7 represents the characters falsely recognized as the target label with high probability. Similarly, for a numeric classifier $f$ trained in the mutually exclusive setting of Table 2, table 8 represents the characters falsely recognized as the target label with high probability.

In the exactly/partly same setting, we see in Table 7 that not many letters are falsely recognized as non-target characters. This results support the fact that the C2G attack against numeric characters works successfully in the exactly/partly same setting (see Figure 4).

On the other hand, in the mutually exclusive setting, Table 8 reveals that many non-target characters are falsely recognized as the target characters. In Figure 4, generated images of "6", "7" and "8" look like "h", "T" and "e", respectively. In Table 8, images of "h", "T" and "e" is falsely recognized as "6", "7" and "8", respectively. From this analysis, if the classifier falsely recognized non-target images as the target label, the C2G attack fails and PreImageGAN generates non-target images as target images.

In Figure 4, images of 0,1,5 and 9 seem to be generated successfully while images for the other numeric characters are more or less disfigured. In Table 8, "O", "l", "s" and "q" are falsely recognized as "0", "1" ,"5" and "9", respectively. This suggests that $f$ does not necessarily contain appropriate features to recognize the target. However, the C2G attack could generate images that are quite similar to the target characters using $f$, and eventually, the C2G attack generated images look like "0", "1", "5" and "9" successfully. For the remaining characters, since $f$ does not contain the necessary information to generate the images of the target, the resulting images are disfigured.

These results suggest that, for the C2G attack to successfully reconstruct a target character, non-target characters should not be falsely recognized as the target character.

Table 7: Top-five non-target characters falsely recognized as target characters (numeric) by an alphanumeric (upper/lower-case numeric) classifier. "(X: 0.123)" means that the classifier misclassified 12.3%(0.123) of images of "X" as the target character with high probability ($>0.9$).

| misclassified as | Top-5 falsely recognized characters | | | | |
|---|---|---|---|---|---|
| 0 | | | | | |
| 1 | (l: 0.028) | (i: 0.020) | (I: 0.005) | | |
| 2 | (Z: 0.102) | (z: 0.054) | (a: 0.024) | (3: 0.002) | |
| 3 | (z: 0.018) | | | | |
| 4 | (y: 0.035) | | | | |
| 5 | (8: 0.002) | | | | |
| 6 | (G: 0.044) | (b: 0.033) | | | |
| 7 | (T: 0.013) | | | | |
| 8 | (q: 0.016) | (g: 0.016) | (P: 0.007) | (S: 0.003) | (2: 0.002) |
| 9 | (q: 0.197) | (g: 0.062) | | | |

Table 8: Top-five non-target characters falsely recognized as target characters (numeric) with high probability by numeric classifier. "(X: 0.123)" means that the classifier misclassified 12.3%(0.123) of images of "X" as the target character with high probability ($>0.9$).

| misclassified as | Top-5 falsely recognized characters | | | | |
|---|---|---|---|---|---|
| 0 | (O: 0.946) | (o: 0.851) | (D: 0.493) | (Q: 0.362) | (c: 0.200) |
| 1 | (l: 0.904) | (I: 0.744) | (i: 0.653) | (L: 0.052) | (j: 0.029) |
| 2 | (Z: 0.878) | (z: 0.786) | (d: 0.462) | (a: 0.201) | (N: 0.094) |
| 3 | (j: 0.147) | (J: 0.141) | (B: 0.080) | (D: 0.045) | (S: 0.042) |
| 4 | (H: 0.896) | (y: 0.491) | (v: 0.423) | (Y: 0.360) | (V: 0.271) |
| 5 | (s: 0.600) | (S: 0.549) | (J: 0.282) | (E: 0.059) | (F: 0.042) |
| 6 | (b: 0.670) | (G: 0.400) | (h: 0.176) | (L: 0.130) | (k: 0.098) |
| 7 | (T: 0.333) | (m: 0.075) | (p: 0.044) | (P: 0.040) | (Y: 0.040) |
| 8 | (B: 0.373) | (R: 0.333) | (x: 0.293) | (X: 0.275) | (e: 0.246) |
| 9 | (q: 0.574) | (g: 0.484) | (Q: 0.106) | (P: 0.093) | (p: 0.089) |

## C    INTERPOLATION ABILITY OF PREIMAGEGAN

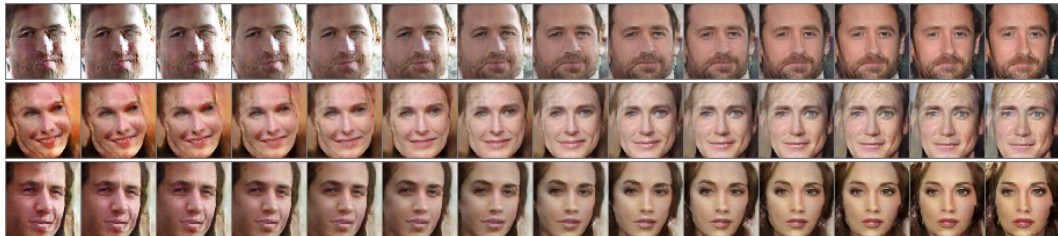

(a) Interpolation on both **z** and **y**. **z** and **y** is randomly chosen from $d_Z$ and $d_{Y_{aux}}$, respectively.

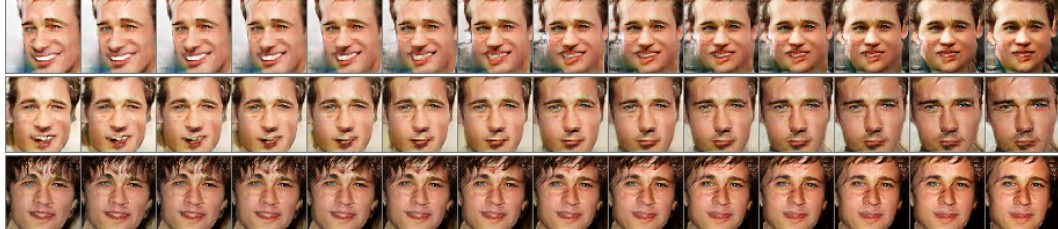

(b) Interpolation on **z** with fixing **y**. **y** is set to one-hot vector in which the element corresponds to "Blad Pitt" is activated.

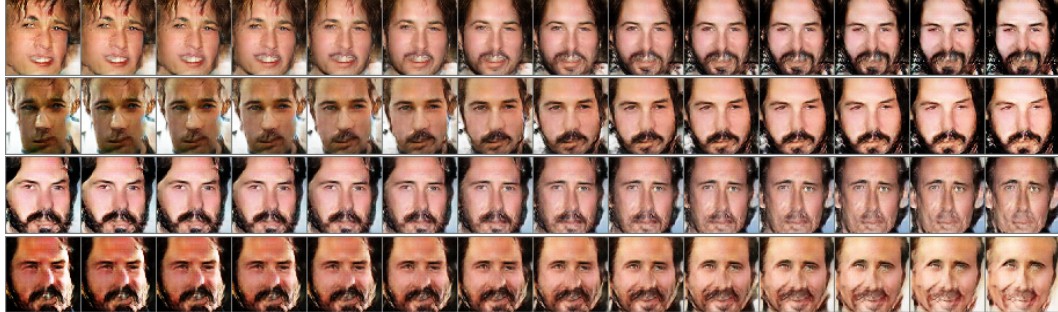

(c) Interpolation on **y** with fixing **z**.

Figure 8: Interpolated images using PreImageGAN.

## D    FACESCRUB EXPERIMENT FULLNAMES

We employed the following 100 people out of 530 people in the FaceScrub dataset for the target. The images of the remaining 430 people are used for the auxiliary dataset. For the full list of the 530 people, please refer to Ng & Winkler (2014).

Matt Damon, Jim Carrey, Tom Hanks, Al Pacino, Brad Pitt, Clint Eastwood, Harrison Ford, Colin Firth, Bruce Willis, Arnold Schwarzenegger, Christopher Lloyd, Danny Trejo, Denzel Washington, Edi Gathegi, Ethan Hawke, Freddy Prinze Jr., Freddy Rodrguez, George Clooney, Gary Dourdan, Hugh Grant, Ian McKellen, Ioan Gruffudd, Jack Nicholson, Jackie Chan, Jean Reno, Jet Li, John Travolta, Keanu Reeves, Ken Watanabe, Leonardo DiCaprio, Laurence Fishburne, Matthew Gray Gubler, Nicolas Cage, Norman Reedus, Omid Djalili, Owen Wilson, Paul Bettany, Paul Walker, Ray Stevenson, Richard Schiff, Robert Di Niro, Samuel L. Jackson, Shia LaBeouf, Victor Garber, Woody Allen, T.J. Thyne, Matt Dillon, Amaury Nolasco, Bill Cosby, Hector Elizondo, Adrianne Len, Allison Janney, Bobbie Eakes, Bonnie Franklin, Calista Flockhart, Carmen Electra, Cathy Lee Crosby, Dana Delany, Debi Mazar, Edie Falco, Eva Longoria, Faith Ford, Fran Drescher, Gates Mc-Fadden, Gillian Anderson, Heather Locklear, Holly Marie Combs, Ilene Kristen, Jamie Lee Curtis, Jamie Luner, Kaley Cuoco, Kim Fields, Kristin Davis, Lacey Chabert, Lauren Holly, Mila Kunis, Melissa Claire Egan, Nadia Bjorlin, Nicole de Boer, Olivia d'Abo, Peri Gilpin, Portia de Rossi, Rachel Dratch, S. Epatha Merkerson, Swoosie Kurtz, Tatyana M. Ali, Tia Carrere, Valerie Cruz, Victoria Justice, Wendie Malick, Yasmine Bleeth, Eliza Dushku, Catherine Bell, Alley Mills, Aisha Hinds, Sara Gilbert, Tamala Jones, Anne Hathaway, Marg Helgenberger, Margaret Cho.

# E  MODEL ARCHITECTURE

## E.1  EMNIST

**EMNIST: Classifier $f$**

**Input Domain**: $\mathbb{X} = [0,1]^{28\times28\times1}$
**Output Domain**: $\mathbb{Y} = \Delta^d$

|  | Kernel size | Stride | Output channels |
|---|---|---|---|
| Convolution | 5x5 | 2x2 | 32 |
| BatchNormalization |  |  |  |
| Relu |  |  |  |
| Convolution | 5x5 | 2x2 | 64 |
| BatchNormalization |  |  |  |
| Relu |  |  |  |
| Flatten |  |  |  |
| Dropout 0.3 |  |  |  |
| Linear |  |  | 2048 |
| Dropout 0.5 |  |  |  |
| Linear |  |  | $d$ |
| Softmax |  |  |  |

**EMNIST: Generator $G$**

**Input Domain**: $\mathbb{Z} = [-1,1]^{128}, \mathbb{Y} = \Delta^d$
**Output Domain**: $\mathbb{X} = [0,1]^{28\times28\times1}$

|  | Kernel size | Stride | Output channels |
|---|---|---|---|
| Concat ($\mathbf{z}$ and $\mathbf{y}$) |  |  |  |
| Linear |  |  | 7x7x256 |
| Reshape |  |  | 256 |
| Deconvolution | 4x4 | 2x2 | 128 |
| LeakyReLU |  |  |  |
| Deconvolution | 4x4 | 2x2 | 64 |
| LeakyReLU |  |  |  |
| Deconvolution | 3x3 | 1x1 | 1 |
| Sigmoid |  |  |  |

**EMNIST: Discriminator $D$**

**Input Domain**: $\mathbb{X} = [0,1]^{28\times28\times1}$
**Output Domain**: $\mathbb{R}$

|  | Kernel size | Stride | Output channels |
|---|---|---|---|
| Convolution | 3x3 | 1x1 | 64 |
| LeakyReLU |  |  |  |
| Convolution | 4x4 | 2x2 | 128 |
| LeakyReLU |  |  |  |
| Convolution | 3x3 | 1x1 | 128 |
| LeakyReLU |  |  |  |
| Convolution | 4x4 | 2x2 | 256 |
| LeakyReLU |  |  |  |
| Convolution | 3x3 | 1x1 | 256 |
| LeakyReLU |  |  |  |
| Flatten |  |  |  |
| Linear |  |  | 1 |

**EMNIST: Alternative Classifier $f'$**

**Input Domain**: $\mathbb{X} = [0,1]^{28 \times 281}$

**Output Domain**: $\mathbb{Y} = \Delta^d$

|  | Kernel size | Resample | Output channels |
|---|---|---|---|
| Convolution | 3x3 | 1x1 | 32 |
| Bottleneck Residual Block | 3x3 |  | 32 |
| Bottleneck Residual Block | 3x3 | Down 2x2 | 64 |
| Bottleneck Residual Block | 3x3 |  | 64 |
| Bottleneck Residual Block | 3x3 | Down 2x2 | 64 |
| BatchNormalization |  |  |  |
| Relu |  |  |  |
| Flatten |  |  |  |
| Dropout 0.3 |  |  |  |
| Linear |  |  | 2048 |
| Dropout 0.5 |  |  |  |
| Linear |  |  | $d$ |
| Softmax |  |  |  |

## E.2 FACESCRUB

**FaceScrub: Classifier $f$**

**Input Domain**: $\mathbb{X} = [-1,1]^{64 \times 64 \times 3}$

**Output Domain**: $\mathbb{Y} = \Delta^d$

|  | Kernel size | Stride | Output channels |
|---|---|---|---|
| Convolution | 5x5 | 2x2 | 32 |
| BatchNormalization |  |  |  |
| Relu |  |  |  |
| Convolution | 5x5 | 2x2 | 64 |
| BatchNormalization |  |  |  |
| Relu |  |  |  |
| Convolution | 5x5 | 2x2 | 128 |
| BatchNormalization |  |  |  |
| Relu |  |  |  |
| Flatten |  |  |  |
| Dropout 0.3 |  |  |  |
| Linear |  |  | 2048 |
| Dropout 0.5 |  |  |  |
| Linear |  |  | $d$ |
| Softmax |  |  |  |

**FaceScrub: Generator $G$**

**Input Domain**: $\mathbb{Z} = [-1,1]^{128}, \mathbb{Y} = \Delta^d$

**Output Domain**: $\mathbb{X} = [-1,1]^{64 \times 64 \times 3}$

|  | Kernel size | Stride | Output channels |
|---|---|---|---|
| Concat ($\mathbf{z}$ and $\mathbf{y}$) |  |  |  |
| Linear |  |  | 4x4x512 |
| Reshape |  |  | 512 |
| Deconvolution | 4x4 | 2x2 | 256 |
| LeakyReLU |  |  |  |
| Deconvolution | 4x4 | 2x2 | 128 |
| LeakyReLU |  |  |  |
| Deconvolution | 4x4 | 2x2 | 64 |
| LeakyReLU |  |  |  |
| Deconvolution | 4x4 | 2x2 | 32 |
| LeakyReLU |  |  |  |
| Deconvolution | 3x3 | 1x1 | 3 |
| tanh |  |  |  |

**FaceScrub: Discriminator** $D$

**Input Domain**: $\mathbb{X} = [-1, 1]^{64 \times 64 \times 3}$

**Output Domain**: $\mathbb{R}$

|  | Kernel size | Stride | Output channels |
|---|---|---|---|
| Convolution | 3x3 | 1x1 | 32 |
| LeakyReLU |  |  |  |
| Convolution | 4x4 | 2x2 | 64 |
| LeakyReLU |  |  |  |
| Convolution | 3x3 | 1x1 | 64 |
| LeakyReLU |  |  |  |
| Convolution | 4x4 | 2x2 | 128 |
| LeakyReLU |  |  |  |
| Convolution | 3x3 | 1x1 | 128 |
| LeakyReLU |  |  |  |
| Convolution | 4x4 | 2x2 | 256 |
| LeakyReLU |  |  |  |
| Convolution | 3x3 | 1x1 | 256 |
| LeakyReLU |  |  |  |
| Convolution | 4x4 | 2x2 | 512 |
| LeakyReLU |  |  |  |
| Convolution | 3x3 | 1x1 | 512 |
| LeakyReLU |  |  |  |
| Flatten |  |  |  |
| Linear |  |  | 1 |

