# OpenReview forum: "Classifier-to-Generator Attack: Estimation of Training Data Distribution from Classifier"
_ICLR.cc/2018/Conference — Reject_

### Official Review · AnonReviewer2 · 2017-11-09
**a nice paper, some details need to be clarified**

**Rating:** 7
**Confidence:** 3

**Review:**

This paper considers a new problem : given a classifier f trained from D_tr and a set of auxillary samples from D_aux, find D_tr conditioned on label t*. Its solution is based on a new GAN: preImageGAN. Three settings of the similarity between auxillary distribution and training distribution is considered: exact same, partly same, mutually exclusive. Experiments show promising results in generating examples from the original training distribution, even in the "mutually exclusive" setting.

Quality:
1. It is unclear to me if the generated distribution in the experiments is similar to the original distribution D_tr given y = t^*, either from inception accuracy or from pictorial illustration. Since we have hold out the training data, perhaps we can measure the distance between the generated distribution and D_tr given y = t^* directly.

2. It would be great if we can provide experiments quantifying the utility of the auxillary examples. For example, when they are completely noise, can we still get sensible generation of images?

3. How does the experimental result of this approach compare with model attack? For example, we can imagine generating labels by e_t^* + epsilon, where epsilon is random noise. If we invert these random labels, do we get a distribution of examples from class t^*?

Clarity:
1. I think the key here is to first generate auxillary labels (as in Figure 2), then solve optimization problem (3) - this causes my confusion at first sight. (My first impression is that all labels, training or auxillary, are one-hot encoding - but this makes no sense since the dimension of f and y_aux does not match.)

Originality: I am not familiar with relevant literature - and I think the GAN formulation here is original.

Significance: I see this as a nice step towards inferring training data from trained classifiers.

---

> ### Author Response · Authors · 2017-12-26
> **Thank reviewer 2**
>
> > 1. It is unclear to me if the generated distribution in the experiments is similar to the original distribution D_tr given y = t^\*, either from inception accuracy or from pictorial illustration.
>
> We agree that it is preferable to demonstrate the performance of the proposed method quantitatively. Unfortunately, in our problem setting, the true generative distribution is unknown, and it is impossible to measure the utility of the resulting generative model. Instead, we employed the inception accuracy as employed in ACGAN.
>
> >Since we have hold out the training data, perhaps we can measure the distance between the generated distribution and D_tr given y = t^\* directly.
>
> I guess the method you suggested is to measure the divergence between the model obtained by the PreImageGAN and the GAN model learned from the training dataset. Even when we have a holdout dataset, it would be difficult to measure the difference between two GANs because GANs often do not give densities or likelihoods. This topic itself is an attractive future direction but is out of the scope of this study.
>
>
> >2. It would be great if we can provide experiments quantifying the utility of the auxiliary examples. For example, when they are completely noise, can we still get sensible generation of images?
>
> To see how much the quality of the auxiliary dataset affects to the generated images, we tried the C2G attack with using meaningless auxiliary images, that is, uniform noise images.
> As clearly shown in the results (Figure 7), meaningless auxiliary images cannot give meaningful results.
> From these results, we could experimentally confirm that the auxiliary dataset affects the quality of the resulting images significantly.
>
>
> >3. How does the experimental result of this approach compare with model attack? For example, we can imagine generating labels by e_t^\* + epsilon, where epsilon is random noise. If we invert these random labels, do we get a distribution of examples from class t^\*?
>
> We guess the reviewer mentioned the model inversion attack [A,B].
> If the model is shallow as already tried in [A] and [B], model inversion will give the distribution of the target images as suggested by the reviewer.
> However, unfortunately, model inversion does not work with deep architecture as already tested by [C].
> This is a major motivation that we designed the C2G attack using PreImageGAN.
>
>
> [A] Matt Fredrikson, Somesh Jha, and Thomas Ristenpart. Model inversion attacks that exploit confidence information and basic countermeasures. In Proceedings of the 22Nd ACM SIGSAC Conference on Computer and Communications Security, CCS ’15, pp. 1322–1333, New York, NY, USA, 2015. ACM. ISBN 978-1-4503-3832-5. doi: 10.1145/2810103.2813677. URL http://doi.acm.org/10.1145/2810103.2813677.
> [B] Matthew Fredrikson, Eric Lantz, Somesh Jha, Simon Lin, David Page, and Thomas Ristenpart. Privacy in pharmacogenetics: An end-to-end case study of personalized warfarin dosing. In 23rd USENIX Security Symposium (USENIX Security 14), pp. 17–32, San Diego, CA, 2014. USENIX Association. ISBN 978-1-931971-15-7.
> [C] Briland Hitaj, Giuseppe Ateniese, and Fernando Pérez-Cruz. Deep models under the GAN: information leakage from collaborative deep learning. CoRR, abs/1702.07464, 2017. URL http://arxiv.org/abs/1702.07464.

---

### Official Review · AnonReviewer3 · 2017-11-25
**Need more ablation study to clarify the contribution**

**Rating:** 4
**Confidence:** 3

**Review:**

This paper proposed to learn a generative GAN model that generates the training data from the labels, given that only the black-box mapping $f$ from data to label is available, as well as an aux dataset that might and might not overlap with the training set. This approach can be regarded as a transfer learning version of ACGAN that generates data conditioned on its label.

Overall I feel it unclear to judge whether this paper has made substantial contributions. The performance critically relies on the structure of aux dataset and how the supervised model $f$ interacts with it. It would be great if the author could show how the aux dataset is partitioned according to the function $f, and what is the representative sample from aux dataset that maximizes a given class label. In Fig. 4, the face of Leonardo DiCaprio was reconstructed successfully, but is that because in the aux dataset there are other identities who look very similar to him and is classified as Leonardo, or it is because GAN has the magic to stitch characteristics of different face identities together?  Given the current version of the paper, it is not clear at all. From the results on EMNIST when the aux set and the training set are disjoint, the proposed model simply picks the most similar shapes as GAN generation, and is not that interesting. In summary, a lot of ablation experiments are needed for readers to understand the proposed method better.

The writing is ok but a bit redundant. For example, Eqn. 1 (and Eqn. 2) which shows the overall distribution of the training samples (and aux samples) as a linear combinations of the samples at each class, are not involved in the method. Do we really need Eqn. 1 and 2?

---

> ### Author Response · Authors · 2017-12-26
> **Thank reviewer 3**
>
> >In Fig. 4, the face of Leonardo DiCaprio was reconstructed successfully, but is that because in the aux dataset there are other identities who look very similar to him and is classified as Leonardo, or it is because GAN has the magic to stitch characteristics of different face identities together?
>
> To show that image generation by PreImageGAN is NOT a naive cherry-picking of image pieces in the auxiliary images, we conducted the following two experiments.
> First, to show that the auxiliary images do not contain face images that look very similar to the target person (say, Keanu Reeves), we evaluated the probability that each image in the auxiliary dataset is recognized as the target in Figure 6. In Figure 6, only a few images are classified as Keanu Reeves with prob>0.8, while most of the images generated by PreImageGAN are recognized as Keanu Reeves with prob>0.95. This indicates that PreImageGAN can generate images recognized as the target only from auxiliary images that are not recognized as the target.
> Second, to show that PreImageGAN used the "magic" to stitch characteristics of different face identities, we generated images of interpolation between two people in Figure 8. As shown in the figures, the faces are smoothly changed from one person to another.
> This indicates that PreImageGAN is NOT a naive cherry-picking of image pieces in the auxiliary images.
>
>
> >From the results on EMNIST when the aux set and the training set are disjoint, the proposed model simply picks the most similar shapes as GAN generation, and is not that interesting.
>
> We agree that the C2G attack targeting numeric characters in the mutually exclusive (disjoint) setting (Figure 4) does not successfully work for some characters.
> We investigated the reason experimentally in detail and found that the PreImageGAN cannot correctly generate images of the target if the classifier recognizes non-target images as a target. For example, in Figure 4, images targeting "7" look "T". This is because the given classifier recognizes images of "T" as "7" falsely (Table 8). As long as the given classifier recognizes non-target images as target images falsely, the C2G attack cannot correctly reconstruct target images.
> In contrast, if the classifier recognizes target images as the target, and at the same time, the classifier recognizes non-target images as non-target (Table 6), the C2G attack can generate images of the target successfully even in the mutually exclusive setting (Figure 3).
> In the revised version, we added these points in Section 5.2 and Appendix B.

---

### Official Review · AnonReviewer1 · 2017-12-03
**Nice idea, but still need some work**

**Rating:** 4
**Confidence:** 3

**Review:**

The paper proposes the use of a GAN to learn the distribution of image classes from an existing classifier, that is a nice and straightforward idea. From the point of view of forensic analysis of a classifier, it supposes a more principled strategy than a brute force attack based on the classification of a database and some conditional density estimation of some intermediate image features. Unfortunately, the experiments are inconclusive.

Quality: The key question of the proposed scheme is the role of the auxiliary dataset. In the EMNIST experiment, the results for the “exact same” and “partly same” situations are good, but it seems that for the “mutually exclusive” situation the generated samples look like letters, not numbers, and raises the question on the interpolation ability of the generator. In the FaceScrub experiment is even more difficult to interpret the results, basically because we do not even know the full list of person identities. It seems that generated images contain only parts of the auxiliary images related to the most discriminative features of the given classifier. Does this imply that the GAN models a biased probability distribution of the image class? What is the result when the auxiliary dataset comes from a different kind of images? Due to the difficulty of evaluating GAN results, more experiments are needed to determine the quality and significance of this work.

Clarity: The paper is well structured and written, but Sections 1-4 could be significantly shorter to leave more space to additional and more conclusive experiments. Some typos on Appendix A should be corrected.

Originality: the paper is based on a very smart and interesting idea and a straightforward use of GANs.

Significance: If additional simulations confirm the author’s claims, this work can represent a significant contribution to the forensic analysis of discriminative classifiers.

---

> ### Author Response · Authors · 2017-12-26
> **Thank reviewer 1**
>
> >In the EMNIST experiment, the results for the “exact same” and “partly same” situations are good, but it seems that for the “mutually exclusive” situation the generated samples look like letters, not numbers, and raises the question on the interpolation ability of the generator.
>
> We agree that the C2G attack targeting numeric characters in the mutually exclusive setting (Figure 4) does not successfully work for some characters.
> We investigated the reason experimentally in detail and found that the PreImageGAN cannot correctly generate images of the target if the classifier recognizes non-target images as a target. For example, in Figure 4, images targeting "7" look "T". This is because the given classifier recognizes images of "T" as "7" falsely (Table 8). As long as the given classifier recognizes non-target images as target images falsely, the C2G attack cannot correctly reconstruct target images.
> In contrast, if the classifier recognizes target images as the target, and at the same time, the classifier recognizes non-target images as non-target (Table 6), the C2G attack can generate images of the target successfully even in the mutually exclusive setting (Figure 3).
> In the revised version, we added these points in Section 5.2 and Appendix B.
>
> >In the FaceScrub experiment is even more difficult to interpret the results, basically because we do not even know the full list of person identities.
>
> We add the list of person identities used for the training dataset and auxiliary dataset in Appendix E.
>
> >It seems that generated images contain only parts of the auxiliary images related to the most discriminative features of the given classifier.
>
> We do not argue that PreImageGAN makes use of "the auxiliary images related to the most discriminative features of the given classifier" because this is the only clue that the adversary can exploit. To show that image generation by PreImageGAN is NOT a naive cherry-picking of image pieces in the auxiliary images, we conducted the following two experiments.
> First, to show that the auxiliary images do not contain face images that exactly look like the target person (say, Keanu Reeves), we evaluated the probability that each image in the auxiliary dataset is recognized as the target. In Figure 6, we see that only a few images are classified as Keanu Reeves with prob>0.8, while most of the images generated by PreImageGAN are recognized as Keanu Reeves with prob>0.95. This indicates that PreImageGAN can generate images recognized as the target only from the given classification model and auxiliary images that are not recognized as the target.
> Second, to demonstrate that generated images are  NOT a naive cherry-picking of image pieces in the auxiliary images, we generated images of interpolation between two people in Figure 8. As shown in the figures, the faces are smoothly changed from one person to another.
>
> >What is the result when the auxiliary dataset comes from a different kind of images?
>
> To see how much the quality of the auxiliary dataset affects to the generated images, we tried the C2G attack with using meaningless auxiliary images, that is, uniform noise images.
> As clearly shown in the results (Figure 7), meaningless auxiliary images cannot give meaningful results.
> From these results, we could experimentally confirm that the auxiliary dataset affects the quality of the resulting images significantly.

---

### Author Response · Authors · 2017-12-26
**Changes in revised paper**

We thank all the reviewers for giving important comments and discussions to improve our manuscript.
According to the comments, we revised our manuscript as follows:

- We added a new experiment on EMNIST to investigate the behavior of the C2G attack in the mutually exclusive setting (Figure 3 and Figure 4. Figure 3 is new). Additional experiments reveal the situation that the C2G attack succeeds and fails (Table 5 - Table 8).

- To show that the C2G attack is NOT a simple cherry-picking of images similar to the targets from the auxiliary dataset, we added the following two new results:
  (1) We measured the probability with which each image in the auxiliary data is recognized as the targets. The probabilities with which images in the auxiliary data are recognized as the targets are low while images generated by the C2G attack are recognized as the target with very high probability (Figure 6). This indicates that the C2G attack can generate images recognized as the targets using images not recognized as the targets.
  (2) We performed the C2G attack with using random images as the auxiliary dataset. The results show that the C2G attack generates meaningless images when the auxiliary dataset consists of meaningless images (Figure 7)

- We investigated the interpolation ability of the generative model obtained by the PreImageGAN (Figure 8). The results showed that the PreImageGAN has a good interpolation ability.

By reflecting comments from reviewers with additional experiments, the paper becomes longer.
In order to keep the consistency of the manuscript before and after revision, we did not change the structure of the manuscript.
If the manuscript is accepted, we would like to shorten the paper (especially in Section 2 and Section 3) so that the entire paper becomes more compact.

---

### Decision · Program_Chairs · 2018-01-29
**ICLR 2018 Conference Acceptance Decision**

**Decision:**

Reject

**Comment:**

This paper addresses the very important problem of ensuring that sensitive training data remain private. It proposes an attack whereby the attacker can reconstruct information about the training data given only the trained classifier and an auxiliary dataset. If done well, such an attack would be a useful contribution that helps make discussion of differential privacy more complete. But as the reviewers pointed out, it's not clear from the paper whether the attack has succeeded. It works only when the auxiliary data is very similar to the training data, and it's not clear if it leaks information about the training set itself, or is just summarizing the auxiliary data. This work doesn't seem quite ready for publication, but could be a strong paper if it's convincingly demonstrated that information about the training set has been leaked.